# Learning How to Separate Fake from Real News: Scalable Digital Tutorials Promoting Students' Civic Online Reasoning

Carl-Anton Werner Axelsson [1,2,*] , Mona Guath [2,3] and Thomas Nygren [2]

1   Department of Information Technology, Uppsala University, 751 05 Uppsala, Sweden
2   Department of Psychology, Uppsala University, 751 42 Uppsala, Sweden; mona.guath@psyk.uu.se (M.G.); thomas.nygren@edu.uu.se (T.N.)
3   Department of Education, Uppsala University, 750 02 Uppsala, Sweden
*   Correspondence: carl-anton.werner.axelsson@it.uu.se

**Abstract:** With the rise of misinformation, there is a great need for scalable educational interventions supporting students' abilities to determine the trustworthiness of digital news. We address this challenge in our study by developing an online intervention tool based on tutorials in civic online reasoning that aims to teach adolescents how to critically assess online information comprising text, videos and images. Our findings from an online intervention with 209 upper secondary students highlight how observational learning and feedback support their ability to read laterally and improve their performance in determining the credibility of digital news and social media posts.

**Keywords:** civic online reasoning; source criticism; lateral reading; observational learning; fact-checking tutorials

## 1. Introduction

With misinformation (we define misinformation as inaccurate, manipulative or false information, including disinformation, which is deliberately designed to mislead people) on the rise, there is a call for scalable educational interventions to support people's ability to navigate news in updated ways [1–3]. The complicated act of effectively using online information is an aspect of digital literacy including not least an ability to evaluate the credibility of information in multimodal formats [4]. Researchers at Stanford University have coined the term *civic online reasoning* to refer to this act of investigating the credibility of information. The method has been developed from professional fact-checking strategies where these professionals investigate who is behind the information, what its pieces of evidences are and what other sources say.

Research, primarily focused on teenagers, highlights how young people struggle to determine credibility of texts, images and videos when these are presented in deceptive ways [5–7]. New technology makes it difficult to debunk fake videos [8] and it may be difficult to separate an authentic image from a misleading image in a tweet [9]. Researchers therefore call for "more intensive digital literacy training models (such as the 'lateral reading' approach used by professional fact-checkers), which could potentially have larger and/or more durable effects" ([3], p. 7). *Lateral reading* refers to the act of corroborating information by exploring multiple independent sources and is contrasted with vertical reading where the reader judges veracity of information only within the given text. In a digital landscape, lateral reading entails opening new tabs in a web browser to perform information searches to investigate the veracity of claims in news or social media posts by finding out what other sources say. In these searches, the fact-checker must also employ click restraint, which means that one should carefully choose which links to follow for further information rather than just making use of the first given search results. Lateral reading using click restraint with the aim to corroborate or debunk information is key for successful civic online reasoning [10–14].

Efforts to educate the public to become more digitally literate have shown some promising results stimulating people to scrutinise misleading headlines on Facebook and WhatsApp [3], pre-bunking against manipulative tweets through an online game [15] and classroom interventions to support civic online reasoning [10,16]. However, these efforts tend to be theory heavy and resource intensive. Interventions are often made in class with lectures and lessons on fact-checking and little possibility for individual feedback. We address this challenge in our study by developing an online digital tool based on civic online reasoning, which aims to teach upper secondary school students how to critically assess online multimodal information. Specifically, we investigate whether tutorials where the students can observe fact-checking at play—coupled with feedback—can serve as simple step-by-step guidelines to improve civic online reasoning.

### 1.1. Civic Online Reasoning

Ku et al. [6] found that news literate individuals are less likely to share news items on social media, leaving less literate individuals vulnerable as they are more likely to share and be exposed to untrustworthy news. To combat misinformation, Lewandowsky et al. [17] have called for an initiative to find technological improvements in the dissemination of news as well as training of readers' critical thinking. Among other things, the authors proposed to draw on journalistic skills to improve media education. Such an effort was initiated by Wineburg and McGrew [11], which led McGrew and collaborators to develop the concept of civic online reasoning in order to boil down online source criticism to a few relevant and concrete strategies [10–14]. Civic online reasoning is an elaboration of three heuristics used by professional historians to evaluate historical texts and images [18,19]: (a) corroboration (i.e., comparing documents), (b) sourcing (i.e., evaluating the document source before reading its text) and (c) contextualisation (i.e., identifying the document frame of reference).

In their study, Wineburg and McGrew [11] invited professional fact-checkers, historians and university students to evaluate websites with information of importance to citizens. The major finding was that the professional fact-checkers outperformed the other groups in their strategies to detect and debunk misinformation. The group of fact-checkers employed *lateral reading*, which means that they immediately began investigating the publisher of the information by opening additional internet browser tabs before they even read the information on the target website. Furthermore, fact-checkers also employed *click restraint*; they carefully chose to proceed to links in search results that were relevant and not necessarily ranked as the top result. Thus, the fact-checkers, in opposition to the other groups, read multiple, relevant sources in order to be able to corroborate and contextualise the information and understand the publisher's motivation.

Often, online information is not only composed of written text but also accompanied by images and videos, particularly when shared through social media. Information consumers, currently and in the future, must therefore be able to critically assess multimodal sources, which requires not only verbal literacy but also visual literacy. What becomes ever more crucial today is the ability to evaluate the credence of visual and verbal content created by skilled communicators.

A picture can better convey a direct meaning than the abstract symbols of verbal communication in the sense that photographs are direct and analogue to the object they portray, in contrast to written words that need to be actively read before they can be understood. The visual system naturally develops as a child matures through the interaction with the environment, but reading text relies on higher cognitive skills. However, when coupled with written text, images can be used in deceptive ways to manipulate the readers' emotions and lend credence to manipulative texts. This contrast was elaborated upon by Messaris [20] arguing that visual images may require greater literacy than words. Therefore, the need for greater literacy arises, given that skilful communicators can easily manipulate their audiences by using emotional or tendentious imagery to augment a message [21]. Imagery, even if true, can be angled to portray a false picture. Video can be manipulated

with selective editing to make the events portrayed become larger than life to sway public opinion [22]. Similarly, statistical data can be presented in graphs in ways that amplify differences to an unrealistic degree [23]. Judgement of images is highly tacit and a viewer often arrives quickly at an interpretation intuitively. It takes mental effort to engage in critical thinking, but visual perception and judgement are immediate and unconscious; hence, inhibition of first impression to act with reason instead needs deliberate practise.

Today, there are tools and methods available to help people act critically when consuming information online. With the help of search engines, information consumers can easily corroborate information accessed online by investigating its sources. Furthermore, there are multiple search engines that allow one to perform reverse image searches to also find the source of visual material. As new technology is developed to create and disseminate news and other media, tools and techniques for finding and corroborating information are also advancing. There is therefore a constant need to update ones digital literacy skills and use digital tools in updated ways [11,17].

### 1.2. Interventions in Civic Online Reasoning

Teaching civic online reasoning has proved to be a challenging task. Previous research on curricular activity to promote civic online reasoning have produced promising but still quite weak results in advancing individuals in digital source criticism and lateral reading. For example, researchers gave university students an instructional module of two 75 min sessions in civic online reasoning [14]. After these modules, the intervention group scored on average less than 3 points out of a total of 10 points on post-intervention tasks in civic online reasoning. McGrew [10] collaborated with a high-school teacher who gave his students two full lessons on lateral reading in a six weeks intervention in civic online reasoning, after which 32% of students made use of lateral reading in the post-intervention tasks. Similarly, McGrew and Byrne [16] in close collaboration with teachers provided high-school students with two full lessons in lateral reading as an intervention. Only 5% of the students could successfully read laterally after the lessons. Kohnen et al. [24] provided middle schoolers with a 90 min workshop in civic online reasoning. After the workshop, the participants were asked to perform concurrent verbalisation while assessing the credibility of a few websites. The study found that the students were able to perform lateral reading but were often led astray by surface details such as names, logos and other imagery, thus deeming misinformation sites as credible.

Despite somewhat successful efforts to teach information consumers civic online reasoning, there are still shortcomings in improving their performance when it comes to corroboration. Lateral reading is a central aspect of civic online reasoning, which highlights the importance of verifying information by corroborating information and using digital resources in updated ways [11]. What the above studies reveal is that much time and effort are required from researchers, teachers and students, unfortunately with only small payoffs in terms of actual evaluation of sources. This lack of application of source evaluation is a well-known issue [6,25]. The question is whether this is a problem of knowledge transfer. Over the years, many experiments relating to learning have shown to be unsuccessful in eliciting expected transfer from learning experiences [26].

In her study, McGrew [10] found improvements in online reasoning in three out of four tasks. The one task that did not show any improvement was the identification of online advertisements. This aspect was never taught explicitly in any of the online reasoning lessons. One can only assume that a transfer effect was expected or at least desired as the subject was not directly addressed in the intervention. Similarly, the authors of the present paper had similar implicit expectations of transfer when initiating research on civic online reasoning tutorials. We conducted an unpublished pilot classroom intervention with upper secondary students. They were given a video tutorial on lateral reading addressing the three essential keys of news credibility: (a) source, (b) evidence and (c) corroboration. The intervention had no effect on the civic online reasoning tasks given as pre and post-test to the students. The tasks comprised: (a) judging whether videos had been manipulated,

(b) identifying online advertisements and, (c) judging the credibility of pseudo-science websites. We understand this as a lack of congruencies between what was taught in the tutorial and the evaluation tasks themselves.

The question arises as to whether we can find less resource-intensive curricular activities that can produce at least the same impact as hour long lessons by allowing students to get closer to the task at hand. The present paper investigates the possibility of using active corroboration exercises of news items, images and videos, providing students with feedback on performance and simple video tutorials, which show how lateral reading of particular items is performed. Classroom interventions to promote lateral reading indicate that students might learn from observing role models who conduct lateral reading and engage with verifying misinformation [14].

### 1.3. Observational Learning and Feedback

In a study on vicarious learning, Bandura et al. [27] found that young children were influenced by observing an agent and imitated its social behaviour if the behaviour was rewarded. Learning through observing experts or role models is a common method when developing practical skills and is routinely employed in, for example, sports [28] and medicine [29,30]. However, the method has also been applied in traditional classroom settings in subjects such as mathematics [31], reading [32] and writing [33]. Collins et al. [34] termed the classroom use of observational learning *cognitive apprenticeship*, thus viewing observational learning in the tradition of craft apprenticeship where a novice works alongside an expert to acquire new skills. Nevertheless, instead of learning practical skills, the cognitive apprentice is taught how to think and reason as an expert and practise mental skills.

Numerous studies have shown the success of giving students worked examples and the importance of giving students more than one example of how to solve a task-specific problem [35]. Learning by observing others or being provided with examples of solving problems can in fact be more helpful for learning than plain reinforcement [36]. Bandura's work has since been elaborated into a social cognitive model of learning divided into phases of observation, emulation, self-control and self-regulation [37]. The idea is that the learner starts emulating the behaviour of the observed model and, through feedback, the learner can then apply the skill independently.

In the learning process, a student benefits most from formative feedback, especially when it is elaborative [38]. Formative feedback is information relayed to the learner during or between instances of task performance in order to shape behaviour and learning towards an effective performance. Hattie and Timperley [39] make a distinction between feedback of task performance and task processing. Task performance feedback is a type of corrective feedback, providing the learner with feedback on how well a particular task was performed, and it can enhance self-efficacy, which, in turn, leads to adaptation of better strategies [39]. Task processing feedback relates to performance strategies and error detection. If students can identify their errors, they are more likely to find more effective strategies and perform better information searches as it increases the likelihood of the student identifying vital cues.

Elaborative feedback means providing the learner with relevant cues or strategies to achieve a correct answer or other desired goals. The most effective feedback tends to be that which gives information about the task and how it should be performed more effectively in relation to goals and in the form of video-, audio-, or computer-assisted instructions [39]. Elaborative feedback can be facilitative, using worked examples or guidance. Elaborative feedback is contrasted with verificatory feedback of only stating whether a response or behaviour was correct or incorrect. However, the most effective feedback contains both elaborative and verificatory information [38]. Verification can be provided explicitly by the student being told whether a response is correct or incorrect, but it can also occur implicitly when a student encounters an unexpected result during performance. It has been found that feedback provides the strongest effect when unsuspecting learners find out they are incorrect [40].

### 1.4. Present Study

The research of the present paper is an attempt to develop simple tutorials in civic online reasoning to be incorporated in an online tool to train students in digital civic literacy called *The News Evaluator* (www.nyhetsvarderaren.se, accessed on 23 February 2021 [41,42]). These tutorials serve a novel approach in demonstrating aspects of civic online reasoning, such as lateral reading and click restraint, in video tutorials. Instead of explaining civic online reasoning as a theory, the tutorials focus on demonstrations of click restraint and corroboration through lateral reading of viral news items, videos and images designed to go viral on social media. Students are presented with these items and then provided with tutorials on how to corroborate their validity using fact-checking strategies. The study of the present paper investigates whether these tutorials can serve as simple step-by-step guidelines to improve participants' digital literacy.

Students are given the task of investigating the validity of the items, after which the tutorials serve as implicit facilitative feedback on the task process where the students can update their strategies. In conjunction with the tutorials, the students are also given explicit feedback on task performance. This is done by displaying their answers, together with a written statement on the item regarding how a professional fact-checker might reason about the item, giving ample opportunities for students to revise their strategies and then practice on a new set of items. Two experiments are presented where we investigate whether this relatively simple intervention would make adolescents more likely to adopt civic online reasoning strategies to facilitate better assessment of online information. This tool, consisting of items, tutorials and feedback was developed to enable students to work on their own. It was created for distance education and data were collected online during the Covid-19 pandemic.

## 2. Method: Experiment I

In Experiment I, participants were asked to fill in an online questionnaire comprising background variables, attitudes and self-rated skills prior to taking the pre-test. They were randomly assigned to either the control group (only taking the pre- and post-tests) or the intervention group that was given tutorials and written feedback on the test items before taking the post-test. All data were collected in one and the same session, which lasted approximately 20 min from start to finish. We hypothesised that the intervention would have a statistically significant positive effect on performance on all post-test items as well as a transfer item.

### 2.1. Participants

Ninety upper secondary school students aged 16–19 volunteered to participate in the study during scheduled lessons led by teachers. The sample comprised 60 individuals identifying as female and 28 individuals identifying as male with 49 students in first upper secondary class, 39 secondary upper secondary class and 1 student in third upper secondary class. Two students did not specify gender and one student did not specify year of upper secondary class.

### 2.2. Design

The experiment was a one-way mixed factorial design with intervention as a between-subjects variable and pre/post measurement as a within-subject variable. The intervention (described in detail below) comprised participants being given tutorial and feedback or no tutorial or feedback on 5 fact-checking tasks. Five tasks, and a control task with no tutorial or feedback, were administered before and after the intervention. The dependent variable was the participants' credibility assessment of the fact-checking tasks, which they provided by selecting a point on a continuous scale from 'very unreliable' (0) to 'very reliable' (1000), represented by a track bar (see Figure 1).

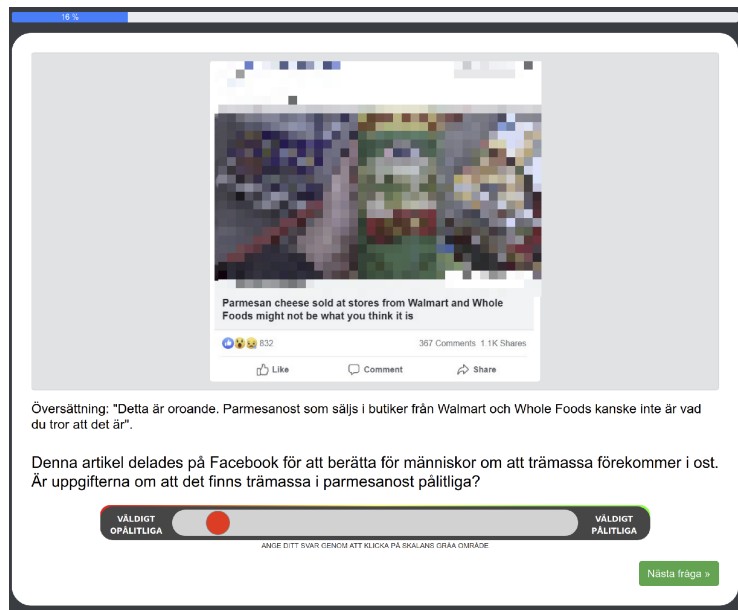

**Figure 1.** Screenshot of an experimental item with track bar allowing participants to provide their credibility assessment

### 2.3. Procedure and Material

At the beginning of the experiment, participants were asked to indicate demographic variables: (a) gender identity, (b) what year of upper secondary education they were in, (c) orientation of education and (d) whether they speak multiple languages at home. Next, participants were asked four questions about attitudes and self-rated skills related to online information. These attitudes and self-rated skills were, however, not analysed in the present study.

The main task comprised 11 unique items, one of which was used in both pre- and post-intervention as a transfer control. In total, each participant assessed 12 items (see Table 1 for a complete description). The non-transfer items were categorised as follows: (a) viral (With the term *viral* we refer to items that are designed to attract attention, being spread and generating clicks.) and true food-related post, (b) manipulated image, (c) fake video, (d) viral and true nature-related post and (e) viral and false post. The transfer control item was a picture of a boy seemingly sleeping between two graves (previously used in [10]). This transfer item was presented both in pre-and post-test and participants were not given instructions on how to assess it. In this way, the item could be used to investigate whether the intervention transferred to a control item.

**Table 1.** Items used pre- and post-intervention.

| Item Type | Pre | Post |
|---|---|---|
| Manipulated image | Photo by Kai Bastard, portraying a woman branded with the words 'You Belong to Me' | Photo by Kai bastard called 'Kiss of Death' showing someone smoking with black veins around the mouth |
| Fake video | Animated Amazon blimp with delivery drones | Animated theme park ride called 'Gyro Drop' |
| Viral false | Post from Eat Local Grown warning about toxic rice from China | Post on the connection between 5G and the Covid-19 virus |
| Viral true, nature | Image shared on Facebook with a one-eyed kitten called Cy | Post from Business Insider of a lake in Australia which turns pink when it rains |
| Viral true, food | Business Insider post on wood pulp being present in Parmesan cheese | Boing Boing post on restaurants using meat glue to produce steaks |
| Viral false, transfer | Post on a picture of a boy seemingly sleeping between his parents graves in Syria | Same post as in pre-test |

Participants were instructed at the beginning of the experiment that they were allowed to use the internet when answering the questions. After having assessed the final item (the control item), participants were asked whether they used any digital aids to assess the items. If they answered 'yes' to this question, they were asked which tools they used (i.e., text search, use of multiple search engines, reverse image search, or specifying other aids). For the experimental group, this question was posed both pre- and post-intervention; for

the control group, the question was asked after the post-test items had been completed. This question was a sign of whether the students used civic online reasoning strategies such as lateral reading to assess the items.

The intervention comprised two parts (see Figure 2) and was administered directly after the pre-test items. First, participants were asked to play a video tutorial showing strategies on how to debunk a specific item from the pre-test. The tutorial was a screen cast produced by one of the authors, showing how an expert would fact-check the specific item. The screen cast was accompanied with a speaker's voice explaining the procedure and how the information could be interpreted. Second, underneath the video, the participant's assessment response was displayed and participants were given a short written statement on how the authors assessed the item and how they came to this conclusion. This was repeated for all five items, but not for the viral and false transfer item.

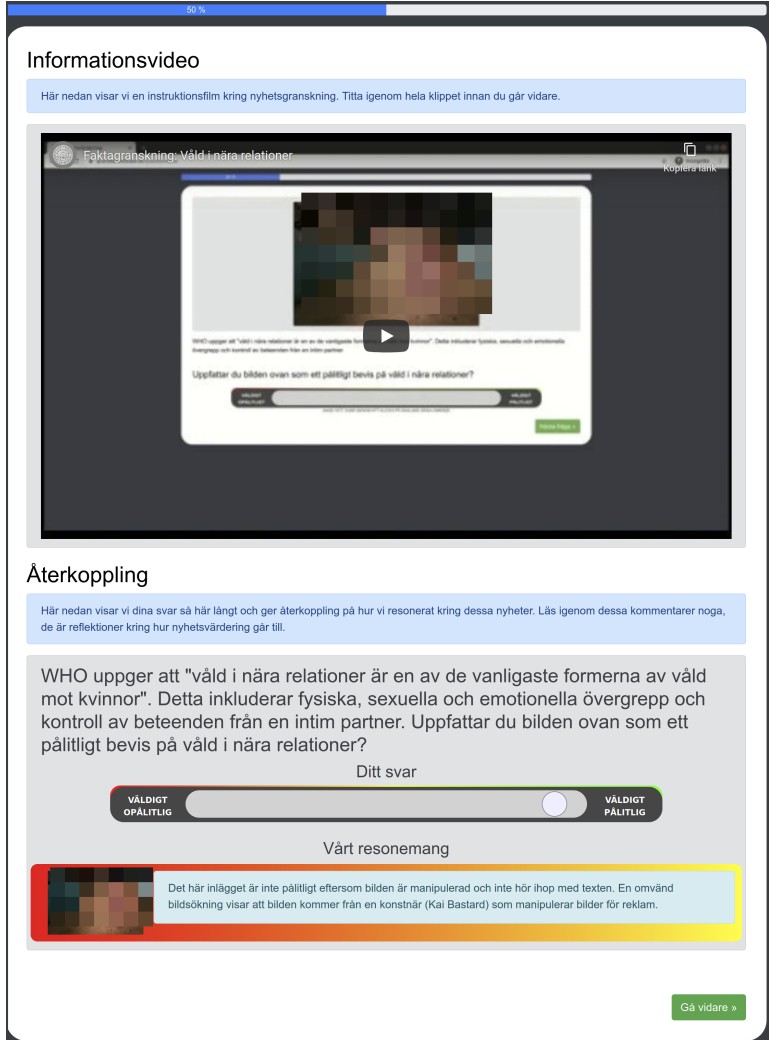

**Figure 2.** Screenshot of the intervention setup where participants were presented with a video tutorial and written feedback.

### 2.4. Analysis

The difference between the pre-test and post-test credibility assessment ratings for the paired items (see Table 1) was analysed by transforming the raw data ranging from 0 to 1000, to a scale ranging from 0 to 1. We then reversed the ratings of the false items so that ratings of all items could be considered a percentage of the correct response. Thereafter, we analysed the mean difference of all items between treatment groups. Next, we made a

more fine-grained analysis for each item. Finally, we analysed the relationship between the use of digital aids, total score and treatment group.

## 3. Results: Experiment I

Figure 3 reveals that the intervention group performed better already in the pre-test, and, importantly, likewise in the post-test. For the control group, the mean of the summed ratings on the pre-test was 0.50 ($SD = 0.09$) and on the post-test 0.58 ($SD = 0.13$). For the intervention group the mean of the summed ratings on pre-test was 0.58 ($SD = 0.12$) and 0.66 ($SD = 0.16$) in the post-test. Interestingly, the control group also improved their total score on the post-test. The distribution of participants using digital aids in the post-test was 7 in the control group ($n = 45$) and 23 in the intervention group ($n = 46$). The difference was statistically significant ($\chi^2(1) = 11.25, p < 0.001$).

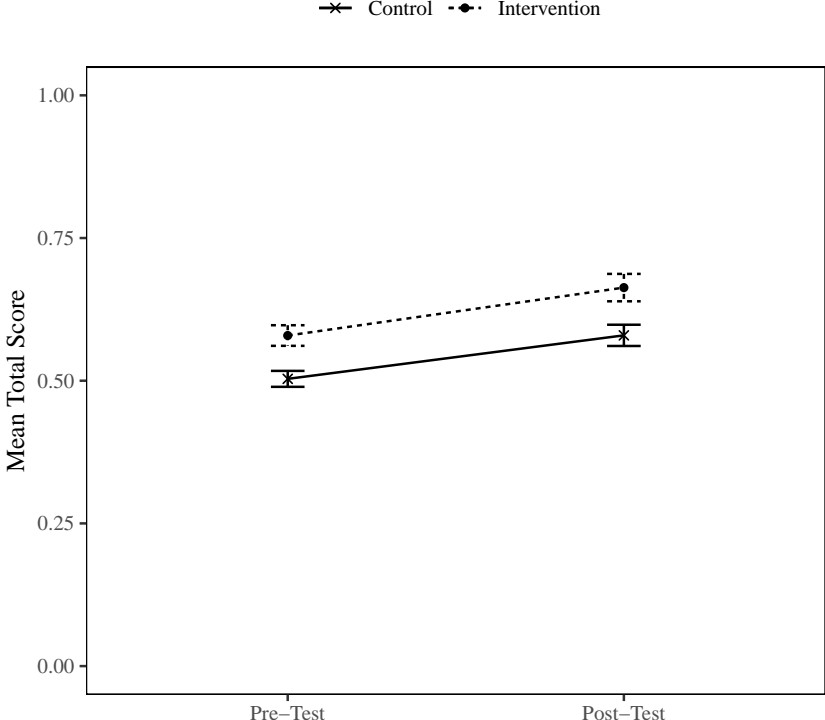

**Figure 3.** Mean total score of all items in the pre- and post-test for the intervention and control group (error bars represent standard error of the mean).

### 3.1. T-Tests on Total Score

In order to investigate whether there were any differences in the total number of correct answers between the groups, we made independent sample Welsh *t*-tests on the difference between the post- and pre-test score. The Welsh *t*-test does not assume equal variance in the groups resulting in fractional degrees of freedom. All the following assumptions were met: (a) independence of observations, (b) no significant outliers and (c) normality. The tests revealed a statistically non-significant difference between groups ($t(85.51) = 0.30, p = 0.77$). Given that both groups improved on the post-test, we did a Welsh *t*-test between the control and intervention group on the post-test score, showing a statistically significant difference ($t(82.87) = 2.76, p = 0.007$) in favour of the intervention group.

### 3.2. Wilcoxon Rank Sum Test on Each Separate Post-Test Item

Since data were not normally distributed in the post-test items, we analysed data with the Wilcoxon rank sum test with continuity correction for independent samples. For the viral and true food post-test item there was a statistically significant difference between the

control and intervention group ($W = 1520, p < 0.001$). The median for the control group was 0.21 ($MAD = 0.19$) and 0.51 ($MAD = 0.37$) for the intervention group; hence, the latter group performed better. For the viral and true nature post-test item, there was a statistically significant difference between the intervention and control group ($W = 1274, p = 0.019$). The median for the control group was 0.09 ($MAD = 0.13$) and 0.23 ($MAD = 0.31$) for the intervention group; hence, the latter group performed better. For post-test items, manipulated image, fake video, viral and false item and, viral and false transfer item, there were no statistically significant differences between the intervention and the control group (all $p > 0.3$).

### 3.3. Relation Between Digital Aids, Score and Treatment

In order to investigate how the use of digital aids was related to total the post-test score and the treatment group, we performed a logistic regression with the use of digital aids (yes/no) as the dependent variable and an interaction between the treatment group and the post-test score. Results revealed an increased log-odds [$b = 8.16, z = 2.93, p = 0.003$] for using digital aids, with a unit increase of the total score. For the separate items on the post-test, there was an increased log odds [$b = 2.42, z = 2.09, p = 0.04$] for using digital aids, with an increase of the manipulated image score. There was a statistically significant effect of the fake video score, amounting to an increased log odds [$b = 3.34, z = 2.87, p = 0.004$] for using digital aids, with a unit increase of the fake video score. This effect was, however, qualified by an interaction (depicted in Figure 4) with the treatment group [$b = -3.37, z = -2.03, p = 0.04$], amounting to a higher log odds for using digital aids with a higher score when being in the intervention group compared with the control group.

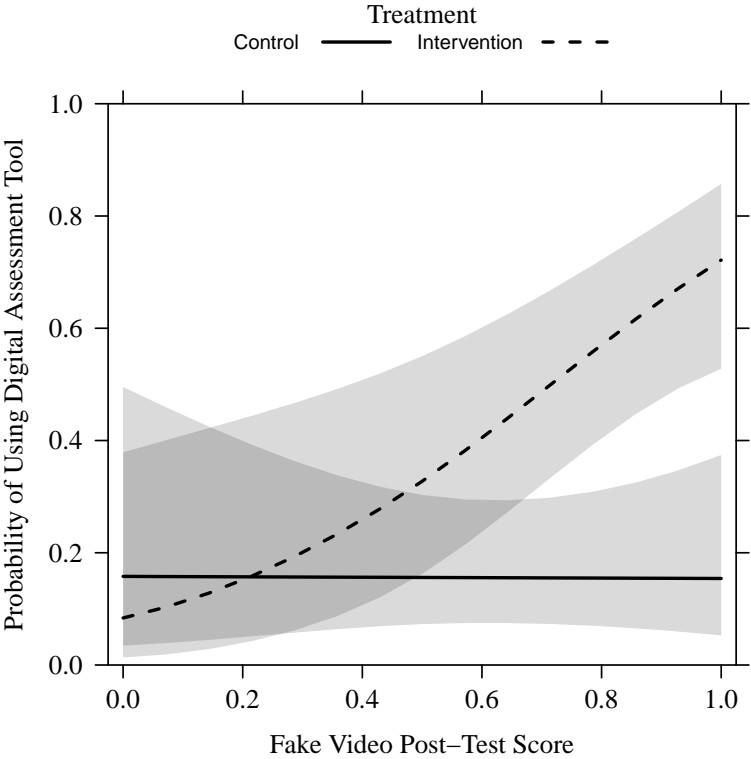

**Figure 4.** Probability of using digital aids as a function of the fake video post-test score and treatment group.

There was an effect of the treatment group on the use of digital aids for the viral and true nature item, amounting to a smaller log odds [$b = -2.20, z = -1.98, p = 0.05$] for using digital aids when being in the control group compared with the intervention group. Finally, there was a statistically significant effect of the score on the viral and false item, amounting to a larger log odds [$b = 3.93, z = 2.09, p = 0.04$] for using digital aids with a unit increase of the score. There were no statistically significant effects of the score or

treatment on the viral and true food item or the viral and false transfer item on the use of digital aids.

## 4. Method: Experiment II

Results from Experiment I showed that merely taking the test improved the performance, which made it difficult to evaluate the impact of the intervention. For this reason, we conducted a second experiment where the control group was given a distraction task in the pre-test; thereafter, the control group was presented with the post-test. The intervention group was given the same treatment as in Experiment I.

### 4.1. Participants

One-hundred and nineteen upper secondary school students agreed to participate in the study during scheduled lessons led by teachers. The sample comprised 50 individuals who identified as female, 68 as male and one individual as nonbinary. Ninety-nine students were in first upper secondary class and 20 were from second upper secondary class.

### 4.2. Design

The experiment was a one-way factorial design with treatment (yes/no) as between-subject variable, where the intervention group was measured on pre- and post-tests, whereas the control group was only measured on the post-test. The dependent variable was the continuous credibility assessment rate of the fact-checking tasks. In order to keep the conditions as similar as possible between the groups, the control group performed a distraction task (described below), while the intervention group did the pre-test. In all other respects, Experiment II was similar to Experiment I (e.g., test items, background measures and attitude/skill measures).

### 4.3. Procedure and Material

As a pre-test, the control group was given a brief definition of artificial intelligence (AI) and asked six questions on their opinions on AI and its societal consequences. Responses were provided by clicking on a track bar (see Figure 5). Similar to the intervention group, the control group was then shown videos and given feedback on their answers (see Figure 6). However, the videos were short clips from a documentary about AI, available from Swedish public service TV (SVT), and not about online fact-checking.

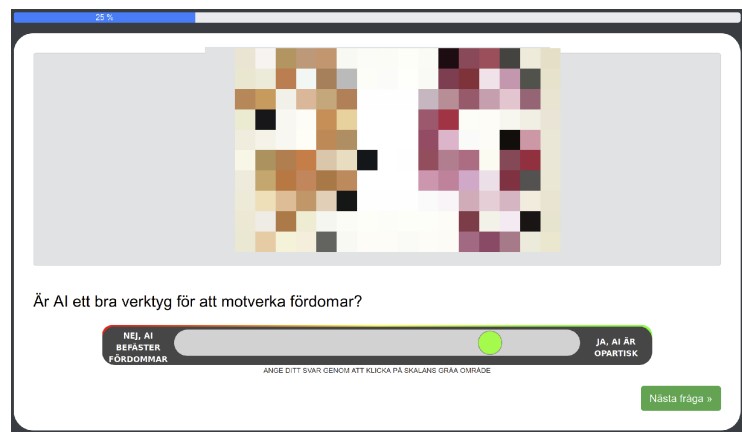

**Figure 5.** Screenshot of one of the questions used in the distraction task of Experiment II.

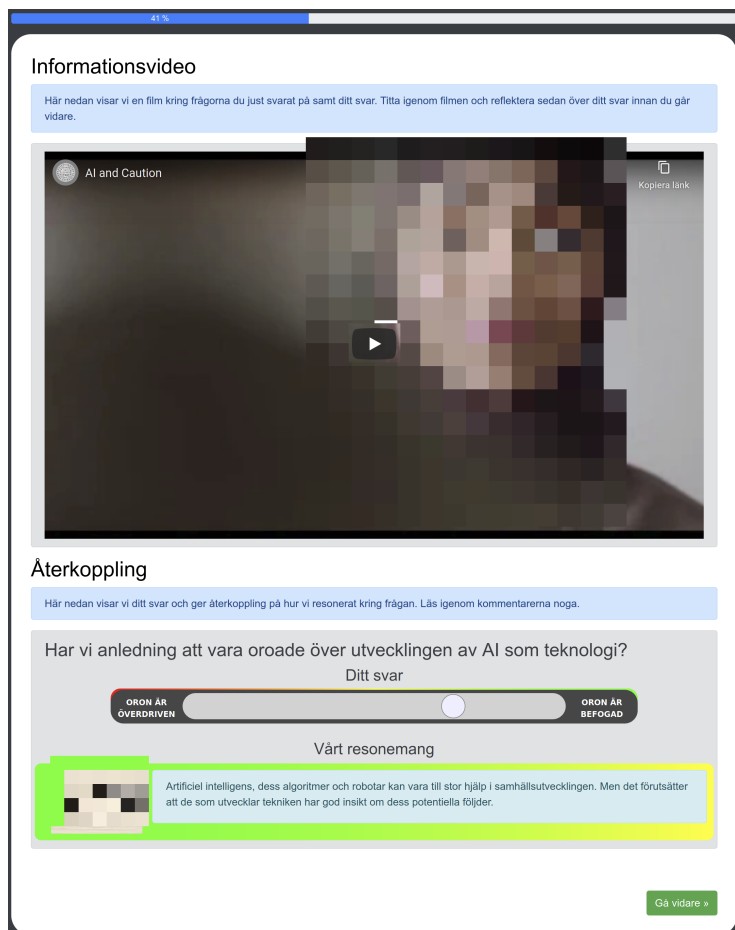

**Figure 6.** Screenshot of the feedback setup for the distraction task of Experiment II.

## 5. Results: Experiment II

We performed the same data analyses in Experiment II as in Experiment I, but only on the post-test scores. Figure 7 reveals that the intervention group performed worse on pre-test than the control group did on the post-test. The figure also reveals a quite large improvement in the intervention group, going from a mean of approximately 0.4 to 0.6 on the total score. For the control group, the mean total score on post-test was 0.57 ($SD = 0.10$). For the intervention group, the mean total score on the pre-test was 0.39 ($SD = 0.12$) and on the post-test 0.60 ($SD = 0.13$). The distribution of participants using digital aids in the post-test was 7 in the control group ($n = 56$) and 30 in the intervention group ($n = 63$). The difference was statistically significant ($\chi^2(1) = 15.47, p < 0.001$).

### 5.1. T-Tests on Total Score

We performed a Welsh *t*-test on the control and intervention group's post-test score, showing a statistically significant difference between the groups ($t(112.6) = 3.59, p < 0.001$), in favour of the intervention group.

### 5.2. Wilcoxon Rank Sum Test on Each Separate Post-Test Item

Since data were not normally distributed in the post-test items, we analysed data with Wilcoxon rank sum test with continuity correction for independent samples. For the viral and true food item there was a statistically significant difference between the control and intervention group ($W = 2586, p < 0.001$). The median for the control group was 0.20 ($MAD = 0.18$) and 0.52 ($MAD = 0.32$) for the intervention group; hence, the latter group performed better. For the manipulated image item there was no statistically significant difference between the groups; the medians were almost identical: 0.12 ($MAD = 0.17$) and 0.08 ($MAD = 0.12$) for the control and intervention groups, respectively. For the fake video

there was also no statistically significant difference, once again, the medians were almost identical: 0.74 ($MAD = 0.39$) and 0.87 ($MAD = 0.20$) for the control and intervention groups, respectively. For the viral and true nature item, there was a statistically significant difference between the intervention and control group ($W = 2238$, $p = 0.012$). The median for the control group was 0.53 ($MAD = 0.34$) and 0.74 ($MAD = 0.30$) for the intervention group; hence, the latter group performed better. For the viral and false item there was no statistically significant difference and once again, the medians were almost identical: 0.93 ($MAD = 0.11$) and 0.91 ($MAD = 0.14$) for the control and intervention groups, respectively. Finally, for the viral and false transfer item, there was no statistically significant difference, with the following medians: 0.72 ($MAD = 0.28$) and 0.83 ($MAD = 0.26$) for the control and intervention groups, respectively.

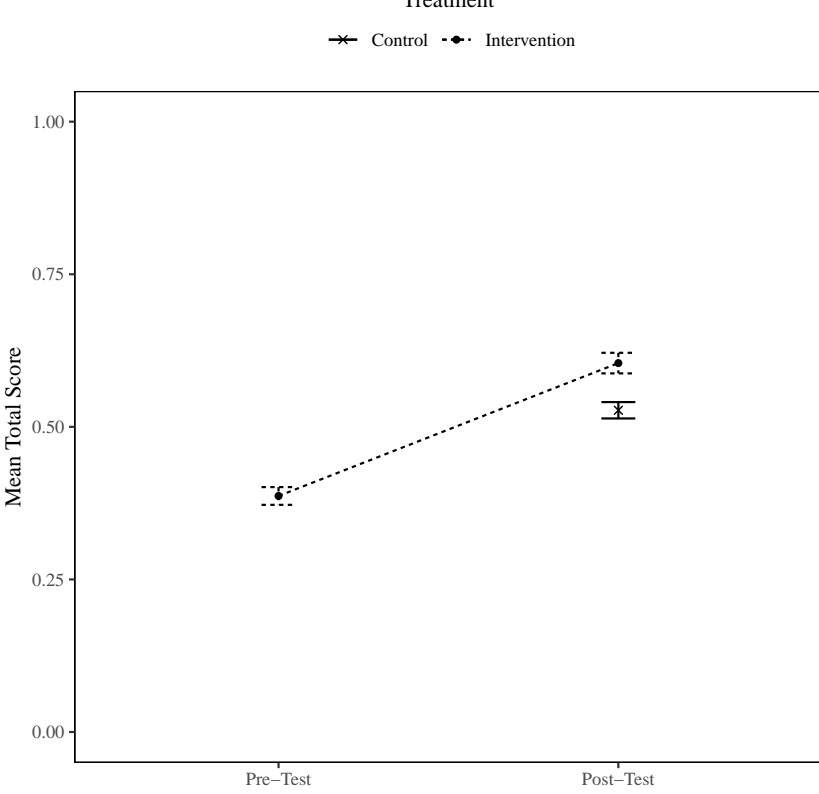

**Figure 7.** Mean total score of all items in pre- and post-test for the intervention group and mean total score of all items in post-test for the control group (error bars represent standard error of the mean).

### 5.3. Relation Between Digital Aids, Scores and Treatment

In order to investigate how the use of digital aids was affected by the total post-test score and intervention group, we performed a logistic regression with the use of digital aids (yes/no) as the dependent variable and an interaction between the intervention group and post-test score. Results showed an increased log-odds [$b = 10.88$, $z = 3.68$, $p < 0.001$] for using digital aids, with a unit increase of the total score.

For the separate items on post-test, there was no statistically significant effect of the viral and true food item score, but there were smaller log odds [$b = -1.80$, $z = -2.10$, $p = 0.04$] for using digital aids when being in the control group compared with the intervention group. There was no effect of manipulated image score, but a smaller log odds [$b = -2.10$, $z = -3.49$, $p < 0.001$] for using digital aids when being in the control group compared with the intervention group. For the fake video score, there was a larger log odds [$b = 7.47$, $z = 3.60$, $p < 0.001$] for using digital aids, with a unit increase of the fake video score. This effect was, however, qualified by an interaction (depicted in Figure 8) between the scores and the treatment group [$b = -6.28$, $z = -2.50$, $p = 0.01$], amounting

to a larger log odds for using digital aids, with a unit increase on the fake video score when being in the intervention group compared with the control group. There were no statistically significant effects of the score or treatment group on the viral true nature item or the viral and false item on the use of digital aids. Finally, there was a statistically significant effect of the score on the viral and false transfer item, amounting to a larger log odds $[b = 3.31, z = 3.05, p = 0.002]$ for using digital aids, with a unit increase of the score on the viral and false transfer item. This effect was, however, qualified by an interaction (depicted in Figure 9) between the score on the viral and false transfer item and the treatment group, amounting to a larger log odds $[b = -3.66, z = -2.07, p = 0.04]$ for using digital aids, with a unit increase of the scores on the viral and false transfer item when being in the intervention group compared with the control group.

### 5.4. Differences between Experiments

Finally, we wanted to compare the performance between the two experiments. First, we calculated Cohen's d with pooled standard deviations for the post-test scores in the treatment and control groups in both experiments. In Experiment I, the effect size of the post-test total score difference was 0.58 and in Experiment II, the effect size of the post-test total score difference was 0.65. Hence, the effect size was somewhat larger in Experiment II, indicating that there was an effect of taking the test in Experiment I. In order to assess whether the effect was statistically significant, we ran a factorial between-subjects ANOVA with the post-test score as the dependent variable and experiment and treatment group as between-subjects variables. Results showed two main effects (depicted in Figure 10): one for experiment ($F(1) = 9.27, MSE = 0.02, \eta_p^2 = 0.043, p = 0.003$) and one for the treatment group ($F(1) = 19.50, MSE = 0.02, \eta_p^2 = 0.087, p < 0.001$). There was, however, no statistically significant interaction ($F(1) = 0.03, MSE = 0.02, \eta_p^2 = 0.0001, p = 0.86$).

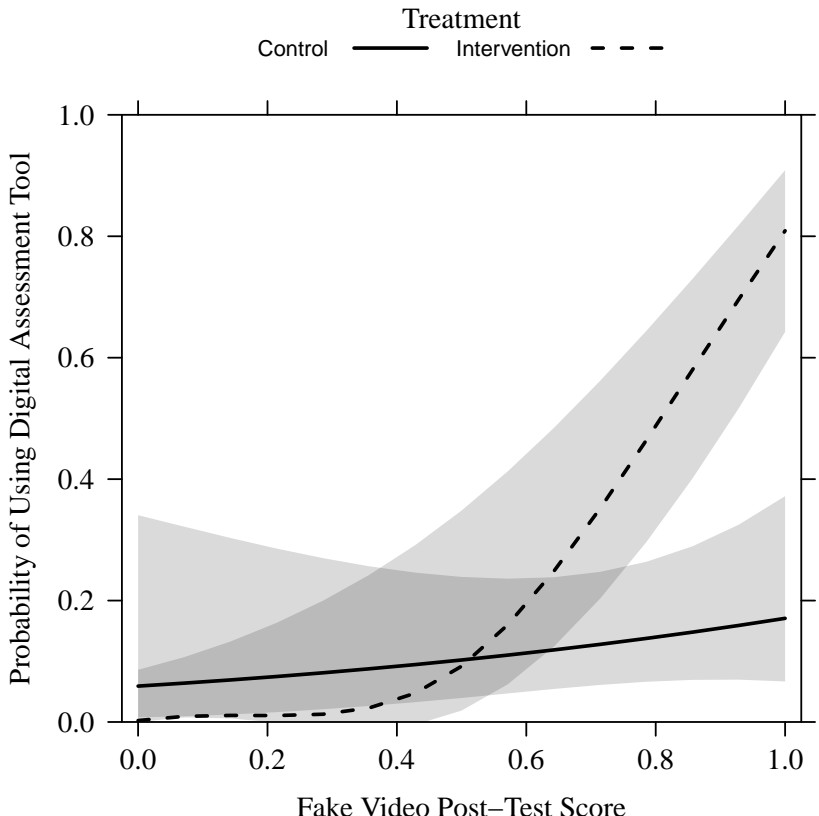

**Figure 8.** Probability of using digital aids as a function of the fake video post-test score and treatment group.

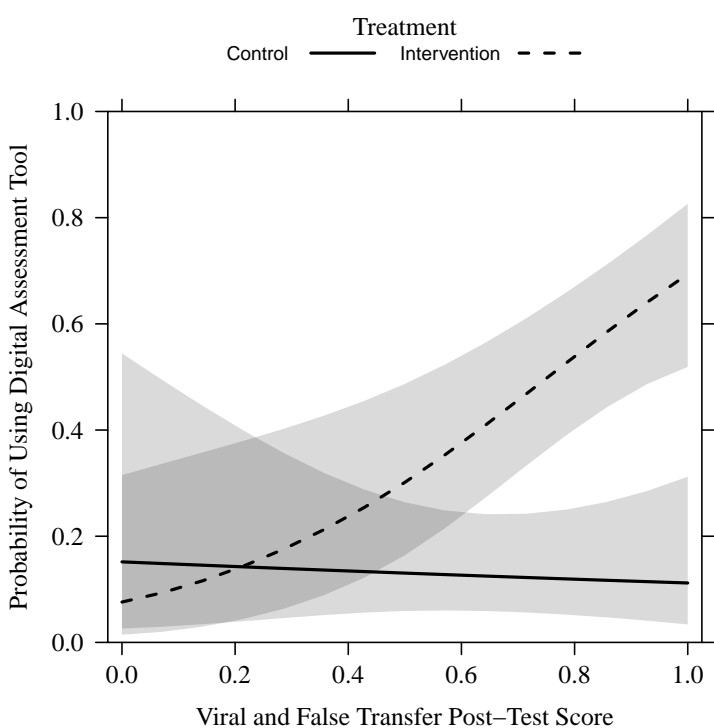

**Figure 9.** Probability of using digital aids as a function of the viral and false transfer item post-test score and treatment group.

In Figure 10 it is clear that participants in Experiment I performed better regardless of the treatment group but that treatment had an effect. A Tukey post-hoc test showed that although participants performed better in Experiment I, there were no statistically significant differences between the control groups with a mean difference ($M = -0.05$, 95% CI $[-0.12, -0.02]$, $p = 0.19$), nor between the treatment groups with a mean difference ($M = -0.06$, 95% CI $[-0.12, -0.007]$, $p = 0.10$).

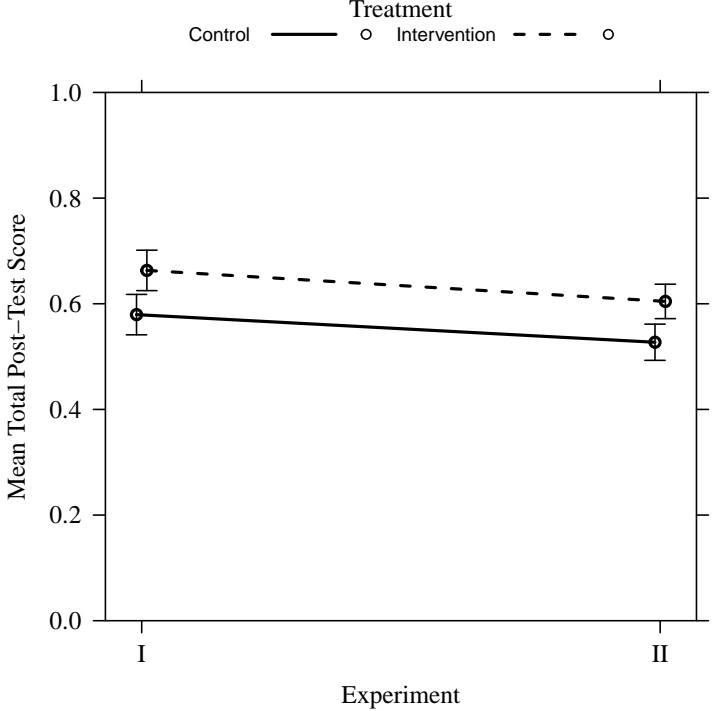

**Figure 10.** Mean post-test score as a function of experiment and treatment group.

## 6. Discussion

In current times, with an abundance of misinformation, consumers of online information must be trained in civic online reasoning to protect democratic rights; thus, citizens need more knowledge on how to verify information [43,44]. With the advancement of information technology, curricular activity in digital literacy must be kept up to date. Researching effective and efficient interventions to support students to use proven techniques and modern tools of fact-checking is therefore paramount [2,3,44]. In the present paper, we have described an online tool developed to train its users in techniques of civic online reasoning to combat misinformation. We created tutorials that show fact-checking hands-on, using typical viral news or misleading posts shared on social media. By letting a student practice fact-checking on these particular items and then presenting the student with feedback on both task performance and task process, the student gains valuable insights on fact-checking strategies. In the present paper, we have presented two experiments where the tool was evaluated with post-intervention tests of new items.

The experiments resulted in three major findings: (a) intervention groups in both experiments exceeded the control groups, in credibility assessment performance of the post-intervention tasks; (b) the intervention increased the likelihood of participants making use of digital aids and use of such tools was related to better performance; and (c) the intervention made participants better at judging unbelievable but true items, suggesting they became more nuanced in their fact-checking. Additionally, the effect sizes of the reported experiments (0.58 and 0.65 for Experiment I and II, respectively) are strong compared to other studies using computer-assisted instructions [45]. We will now summarise the major findings and discuss them in turn.

### 6.1. Better Performance on Post-Tasks for Intervention Group

In our first experiment, we found no statistically significant difference score between the post-test and pre-test. A comparison of treatment effect on the post-test score between the experiments showed a somewhat larger effect size in Experiment II, but it did not translate into statistically significant differences between the treatment groups. The interpretation of the results is obfuscated by the fact that participants performed so differently in Experiment I and II. We therefore conclude that the effect of performing the test, even without feedback, is not negligible. However, more importantly, intervention groups in both experiments outperformed the control groups on post-test tasks. This suggests that the intervention was successful in promoting better assessment results.

### 6.2. Increased Use of Digital Aids

In Experiment I, the probability of using digital aids was positively related to total post-test score and manipulated image score. The probability of using digital aids was also related to the treatment group for the viral true nature item. For fake videos, the relationship was mediated by both scores and treatment group. In Experiment II, the use of digital aids was positively related to scores on viral true nature items and manipulated images. The use of digital aids was also mediated through an interaction between the treatment group and scores and the fake video. There were no statistically significant differences in performance between the control and intervention group in the two experiments with regards to our transfer item. However, we could show that the use of digital aids during the experiment and being in the intervention group increased the assessment score on this item in Experiment II. It should be possible to generalise on such interventions in civic online reasoning and they should prove to be effective on items other than those the students are being trained on. The fact that the intervention led to increased use of digital aids and that such use led to better performance indicates that the tool we have developed has great potential in transferring civic online reasoning strategies to other items. Furthermore, the increased use of digital aids shows that this online intervention tool was successful in stimulating lateral reading strategies. If we can inspire students to

make use of these strategies, then they are more likely to succeed in assessing the validity of online information.

### 6.3. Better Assessment Performance of True Items

In Experiment I, when looking at each item separately, the intervention group was better at two post-test items, indicated by a statistically significant difference on the viral true food item and viral true nature item. In Experiment II, the differences on each separate item showed that superior performance in the intervention group on the total score can be attributed to the viral true items. Here, we observe quite large differences, whereas on the false items and the transfer items, the differences are very small. In a test situation or in curricular activity in online source criticism, students are probably likely to become inherently sceptical of items presented to them. The finding that this online intervention tool increases performance on true items is therefore vital in fostering nuanced fact-checking. Researchers have noted that there may be a risk that truth decay in news may turn into a trust decay towards credible news [46]. Labelling some news as false may add perceived credibility to dubious news not marked as false [2]. Thus, it is important to support students' ability to identify misinformation and credible news.

### 6.4. Limitations

The difficulty in providing evidence of the efficacy of the online intervention tool with respect to the main effects of difference scores between pre- and post-tests in Experiment I might be explained by the items chosen. Four out of six items were false. This means that any individual who is generally sceptical could perform well without much use of any lateral reading strategies. However, the findings that the tool promotes lateral reading strategies and increases the assessment performance of true items indicate that it is an effective tool even with this limitation.

### 6.5. Conclusions

With the experiments presented in the present paper, we have reported promising results of promoting civic online reasoning strategies to adolescents through the use of tutorials in fact-checking. This intervention takes 20 min and will be available online without charge. This scalable intervention is quite efficient in comparison with other researched interventions that require much time from researchers, teachers and students in lengthy curricular activities. With the two experiments presented in the present paper, we have shown that (a) use of digital aids lead to better credibility assessment, (b) use of such aids was more prevalent in the intervention groups, (c) which led these participants to perform better on the post-task items (identifying, e.g., fake videos or misleading posts on social media) and (d) allowed participants to show a more nuanced performance by being better at assessing reliable news items. We therefore conclude that explicit tutorials and implicit feedback can foster civic online reasoning in adolescents if used in a curricular activity.

**Author Contributions:** Conceptualization, T.N.; Data curation, C.-A.W.A. and M.G.; formal analysis, M.G.; funding acquisition, T.N.; investigation, T.N.; methodology, C.-A.W.A. and M.G.; project administration, T.N.; Resources, T.N.; software, C.-A.W.A.; supervision, T.N.; visualization, C.-A.W.A.; writing—original draft, C.-A.W.A.; writing—review and editing, M.G. and T.N. All authors have read and agreed to the published version of the manuscript.

**Funding:** This article is the result of a research project funded by Vinnova (grant 2018-01279).

**Informed Consent Statement:** Informed consent was obtained from all subjects involved in the study.

**Data Availability Statement:** Not applicable, the study does not report any data.

**Conflicts of Interest:** The authors declare no conflict of interest.

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
