# Peer review of "Learning How to Separate Fake from Real News: Scalable Digital Tutorials Promoting Students’ Civic Online Reasoning"

_futureinternet, doi:10.3390/fi13030060_

Round 1
Reviewer 1 Report
1) Mentioned about the lateral reading approach used by fact-checkers and then you straight jumped out to talk about the digital tool - civic online reasoning without preparing the readers for it. You should discuss what is lateral reading and why are you moving to the tool of your choice.
2) The section title - Civic Online Reasoning prepares the readers to read about the topic title but you start talking about the method fact-checkers use. Reorganize the content based on the comment 1 & 2 to flow well.
3) You are also making assumptions that readers know what lateral reasoning is and you do not take time to explain that even briefly.
4) You do not prep me for the section on observational learning and feedback- I do not know why I am reading it and how to relates to your study.
5) I only knew after reading halfway into the paper that the study participants were school students. The use of "participants" in the abstract is misleading.
Author Response
We would like to thank Reviewer I for the very helpful comments to improve the introduction of our paper.
1) The introduction has been reorganised to better relate the terms 'digital literacy', 'civic online reasoning' (COR) and 'lateral reading' to one another. We toned down the use of 'digital literacy' as it is an umbrella term comprising more than just digital source criticism, instead trying to make it clearer that COR is a part of digital literacy. We have also explained lateral reading in the introduction and its relation to corroboration/click restraint which are fundamental to COR. [Lines 11-46]
2) COR is based on professional fact-checker strategies, this has been emphasised in the introduction to make it clearer why we start discussing fact-checkers. [Lines 16-19]
3) Lateral reading is now briefly explained early in the introduction. [Lines 26-35]
4) We have added the text: "Specifically, we investigate whether tutorials where the students can observe fact-checking at play ..." [Lines 44-45] to better prepare the reader that we make use of observational learning as a tool for learning.
5) The abstract has been revised with a focus on adolescents and upper secondary school students
Reviewer 2 Report
The authors developed and evaluated the impact of an online tool for training civic online reasoning. In my opinion, just a few issues need to be "fixed" before publication.
- The authors should spend additional words to better define the term "hands-on" for a broader audience.
- Please provide a clear statement for testing for T-test assumptions.
- Please specify the time-lapse that occurred between pre and post-tests (for replicability issues).
- In Fig. 3 please indicate in the figure and relative caption what "mean total scores" stands for exactly. Figures should be self-explanatory without referring to the MS text.
Author Response
We would like to thank Reviewer II for their comments.
We chose to remove the term 'hands-on' from the paper and title as elaborating on this term would not substantially improve the paper.
We added the following text to clarify our t-test assumptions:
"In order to investigate whether there were any differences in the total number of correct answers between the groups, we made independent sample Welsh t-tests on the difference between post- and pre-test score. The Welsh t-test do not assume equal variance in the groups resulting in fractional degrees of freedom. All the following assumptions were met: (a) independence of observations, (b) no significant outliers and (c) normality."
[Lines 300-305]
There was no time-lapse between pre- and post as these are administered in the same session. To clarify this, we added the following sentence to the method section of Experiment I: "All data were collected in one and the same session which lasted approximately 20 minutes from start to finish."
[Lines 227-228]
We revised the Figure 3 caption to read "Mean total score of all items ..." to make it clearer that the total score refers to all items of the particular test